# Exchange Bias Demonstrated in Bulk Nanocomposites Processed by High-Pressure Torsion

**DOI:** 10.3390/nano13020344

**Published:** 2023-01-14

**Authors:** Michael Zawodzki, Lukas Weissitsch, Heinz Krenn, Stefan Wurster, Andrea Bachmaier

**Affiliations:** 1Erich Schmid Institute of Materials Science of the Austrian Academy of Sciences, 8700 Leoben, Austria; 2Institute of Physics, University of Graz, 8010 Graz, Austria

**Keywords:** severe plastic deformation, high pressure torsion, nanocomposite, superior hardness, microstructural characterization, magnetic properties, hysteresis, exchange bias

## Abstract

Ferromagnetic (Fe or Fe_20_Ni_80_) and antiferromagnetic (NiO) phases were deformed by high-pressure torsion, a severe plastic deformation technique, to manufacture bulk-sized nanocomposites and demonstrate an exchange bias, which has been reported predominantly for bilayer thin films. High-pressure torsion deformation at elevated temperatures proved to be the key to obtaining homogeneous bulk nanocomposites. X-ray diffraction investigations detected nanocrystallinity of the ferromagnetic and antiferromagnetic phases. Furthermore, an additional phase was identified by X-ray diffraction, which formed during deformation at elevated temperatures through the reduction of NiO by Fe. Depending on the initial powder composition of Fe_50_NiO_50_ or Fe_10_Ni_40_NiO_50_ the new phase was magnetite or maghemite, respectively. Magnetometry measurements demonstrated an exchange bias in high-pressure torsion-processed bulk nanocomposites. Additionally, the tailoring of magnetic parameters was demonstrated by the application of different strains or post-process annealing. A correlation between the amount of applied strain and exchange bias was found. The increase of exchange bias through applied strain was related to the microstructural refinement of the nanocomposite. The nanocrystalline maghemite was considered to have a crucial impact on the observed changes of exchange bias through applied strain.

## 1. Introduction

The introduction and application of nanomaterials are indispensable for progress in technological development. The use of nanomaterials is already well advanced and ranges from applications in medicine (targeted drug delivery for cancer treatment) [1], catalysis (optimisation of the nanostructure of the catalyst to improve the selectivity and efficiency of catalytic processes [2], such as the enhanced reduction of volatile organic compounds [3]), gas sensors [4], microelectronics [5] to engineering [6].

Metallic nanomaterials have outstanding physical properties, such as superior strength, superior hardness and enhanced wear resistance, and attracted the attention of engineers to reduce weight to improve efficiency for transportation vehicles [6,7].

The importance of magnetic materials for power efficiency is commonly known. Material scientist focus here on the improvement of the nanostructure of soft- or hard-magnetic materials to enhance the magnetic properties [8,9]. The important role of nanocrystalline microstructure can be demonstrated for hard magnetic materials. Here, highly textured nanocrystals are considered to be the key to pushing the energy product (BH) of a hard magnetic material to its theoretical maximum of $BHmax=1/4\mu_{0}M_{S}^{2}$ ($\mu_{0}$...vacuum permeability and M_S_...saturation magnetisation) [9,10]. An increase in BH of the hard magnetic material leads to a reduction of weight and size of permanent magnet motors/generators and this addresses directly the need for power efficiency [11]. Although synthesis of nanocrystalline material has been challenging, a variety of methods have been established to obtain the desired nanocrystalline state, (e.g., inert gas condensation, electrodeposition, mechanical alloying, crystallisation out of an amorphous state and severe plastic deformation (SPD)) [12]. Regarding synthesis, SPD methods, especially high-pressure torsion (HPT), offer unique advantages. For example, it is possible to process a bulk nanocrystalline sample based on powder blends, which allows for a large variety of phase combinations. With HPT it is possible to synthesise supersaturated solid solutions [13] or influence the magnetic properties of the nanocrystalline sample through the application of strain [14,15,16,17].

The focus of this study was on the use of rare-earth free phases due to their abundant availability. Therefore, the choices for a suitable ferromagnetic (FM) material were limited to Fe, Ni or FeNi-alloys. Fe was chosen, because of the vast base of experience available concerning HPT-deformation [18]. A suitable alternative FM-phase is the γ-Fe_20_Ni_80_-alloy, which possesses lower M_S_ compared to Fe and far larger domain wall width than Fe or Ni. Those two properties of γ-Fe_20_Ni_80_ are regarded as beneficial to the enhancement of exchange bias (H_eb_).

The H_eb_ was first observed by Meiklejohn and Bean on Co-CoO nanoparticles and introduced as ‘new unidirectional anisotropy’ [19]. H_eb_ originates from an FM spin exchange coupling mechanism between the surface spins of adjacent antiferromagnetic (AFM) and FM phases and allows the preservation of an external magnetic field direction through the alignment of AFM surface spins during field cooling (FC) below the Neél temperature (T_N_). This phenomenon causes a biased shift of the hysteresis in the FC direction.

A further aim of this study was to use a material, which exhibits AFM properties at room temperature (RT). NiO has the benefit of possessing a face-centred cubic crystal structure with a smaller lattice parameter mismatch to Fe or γ-Fe_20_Ni_80_-alloy compared to other possible AFM materials (e.g., FeS or α-Fe_2_O_3_), and is also AFM far above RT; it was, therefore, the material of choice.

Until recently, such material combinations have mainly been realised by thin film deposition techniques to study the interfacial phenomenon of H_eb_ [20], due to its technical application in magnetic read heads for hard-disc drives [21]. Consequently, previous investigations have been limited to 2D structures. A first successful synthesis by HPT of bulk composites, which possess an H_eb_, has been reported for the Co-NiO system [17].

The aim of this study was to obtain a bulk nanocomposite possessing enhanced magnetic properties, H_eb_ and be able to demonstrate tailoring of magnetic parameters via applied strain. Furthermore, to investigate in an initial trial of the new bulk nanocomposite regarding their nanostructure and the influence of the nanostructure on the magnetic properties.

A material synthesis with a metallic and an oxide phase can be a challenging task, especially for HPT, if a homogeneous microstructure is desired. Although oxide ceramics (i.e., Al_2_O_3_, ZrO_2_, TiO_2_ and Y_2_O_3_) have been processed previously with HPT [22], the present study has investigated for the first time the deformation process in combination with nanostructural characterisation and magnetic characterisation systematically to provide a detailed description of the FM-AFM bulk nanocomposite.

A further aim was to attain deeper insight into the HPT-deformation process of NiO itself and its interaction with the mechanically softer FM phase. The importance of HPT processing at elevated temperatures was demonstrated to synthesise homogeneous bulk nanocomposites. In addition to the tailoring of magnetic properties, an unusually high Vickers microhardness was detected, when the NiO phase was preserved during deformation.

## 2. Materials and Methods

Commercially available powders (Fe-Mateck Fe-99.9% 100 + 200 mesh, Ni-Alfa Aesar Ni 99.9% 100 + 325 mesh, NiO-Alfa Aesar NiO 99.8% 325 mesh) were used. All powder compositions are labelled in –at%, if not otherwise noted. Powders were stored and prepared inside a glove box filled with Ar-atmosphere. For powder consolidation with HPT, an airtight capsule was used to prevent the powder from oxidation. The capsule itself enclosed the HPT anvils, which can move freely only in the axial direction for powder blend compaction. The obtained pellet was used later on in a second step for HPT-deformation Figure A1. The processed sample discs had dimensions of (⌀8×0.6) mm [23]. The process parameters were the following: applied hydrostatic pressure of 6 GPa, rotation speed ω=1.25 min${}^{-1}$ and deformation temperatures (T_def_) of 200–300 °C. The T_def_ was provided by inductive heating of both anvils during the processing, and the temperature was controlled by a pyrometer [18]. Samples deformed at RT were cooled with pressurised air to ensure the temperature stability of the sample. Equivalent von Mises strain (ϵ_vM_) was calculated with $\epsilon_{vM}=\left(2\pi Nr\right)/\left(\sqrt{3}t\right)$ (*N*...applied rotations, *r*...radius and *t*...sample thickness) [23].

Scanning electron microscopy (SEM; LEO-1525 Carl Zeiss GmbH, Oberkochen, Germany) was used in backscattered electron mode (BSE; Model 7426, Oxford Instruments plc, Cambridge, England, United Kingdom) with an acceleration voltage of 20 kV. Energy dispersive X-ray spectroscopy (EDX; XFlash 6–60, Bruker Corporation, Berlin, Germany) analysis with SEM was performed at 5 kV to maximise lateral resolution. To assign the corresponding radial position to the SEM images, the ends of the examined HPT half discs were determined during the SEM examination. *r*∼0 mm was defined as the half distance between both ends, and *r*∼3 mm was determined by the relative distance to *r*∼0 mm. *r*∼3 mm was cross-checked by measuring the relative distance to the end of the disc, which should be 1 mm. Hardness measurements were performed with a Micromet 5104 from Buehler (Düsseldorf, Germany) with an indention load of 500 g along the radial direction every 250 μm.

For X-ray diffraction (XRD), samples were polished on the top side and analysed with a D2 Phaser with a Co-K_α_ source (Bruker Corporation, Billerica, MA, USA). XRD data from synchrotron measurements were collected in transmission mode at Deutsches Elektronen-Synchrotron (DESY, Hamburg, Germany) at the beamline P07B (high energy materials science) with a beam energy of 87.1 keV and a beam size of 0.5×0.5 mm${}^{2}$. All wide-angle X-ray scattering (WAXS) measurements were done in axial direction [16] and the data were gained with a Perkin Elmer XRD 1621 flat panel detector (Waltham, MA, United States). Peak analysis was done with a self-written script in Octave (version 6.2.0) based on the pseudo-Voigt method to determine the integral breadth and coherent scattering domain size (CSDS) [24] via Scherrer-relation for spherical crystallites. As an XRD-reference pattern the American Mineralogist Crystal Structure Database (AMCSD) is used (Fe: AMCSD 0012931, Ni: AMCSD 0012932, NiO: AMCSD 0017028, γ-Fe_2_O_3_: AMCSD 0020516, α-Fe_2_O_3_: AMCSD 0000143 and Fe_3_O_4_: AMCSD 0009109).

The annealing experiments were done within a vacuum furnace (XTube, type-1200, Xerion Advanced Heating, Berlin, Germany). For sample preparation, a deformed HPT disc was cut in quarters. One-quarter of the sample material was used to obtain probing material for magnetometry measurements from the same radial position, and the other quarters for microstructural characterisation. For the annealing experiment itself, one full quarter and one sample for magnetometry measurements were heated up to 450 °C or 550 °C in a vacuum of 10^−6^ mbar or better and kept at the specific temperature for one hour.

Magnetometry samples were prepared with a diamond wire saw and samples were cut out at desired radial position, having a volume of ∼2 mm^3^ or less [16]. Due to the sample dimensions, the applied strain, as mentioned for the magnetometry measurements, is an average value and related to the centre of mass of the measured sample.

Field cooling was performed with the applied magnetic field parallel to the radial direction from above the corresponding Néel-temperature from NiO of TN∼524 K [25] to RT inside a vacuum chamber at 10^−4^ mbar or better. An electromagnet provided a magnetic field of ∼10 kOe. The field cooling was subsequently continued below RT, realised by a superconducting quantum interference device (SQUID) from Quantum Design MPMS-XL-7 (San Diego, CA, USA) from RT to 8 K at 10 kOe, which was further used to determine magnetic hysteresis loops for the Fe_10_Ni_40_NiO_50_ composition. Measurements were executed at 8 K with a maximum magnetic field strength of ±70 kOe to ensure magnetic saturation. After determining H_C1_ and H_C2_, the H_eb_ can be calculated with $H_{eb}=(H_{C1}+H_{C2})/2$. The coercivity of the symmetric hysteresis is determined by $H_{C}=((-)H_{C1}+H_{C2})/2$. M_S_ was approximated with a linear fit from the saturated branches of hysteresis to obtain M_S_ at the intercept with the ordinate.

## 3. Results

### 3.1. Hardness and Microstructural Evolution

#### 3.1.1. Fe_50_NiO_50_ Composition

The first composition investigated was the binary Fe_50_NiO_50_ (Fe_42.8_NiO_57.2_ in -wt%) blend. Figure 1a,b present SEM micrographs at *r*∼0 mm and *r*∼3 mm. Both samples were processed with 100 rotations and the deformation temperature was increased from RT (Figure 1a) to 300 °C (Figure 1b). This yielded an applied strain for T_def_ = RT of ϵ_vM_∼1760 and for T_def_ = 300 °C a ϵ_vM_∼1490 at *r*∼3 mm, due to a slight height difference. For the sample processed at RT, EDX measurements confirmed that the dark phase seen in BSE contrast consisted of NiO, while the bright phase was identified as Fe-phase.

When processed at RT, both phases formed lamella structures (Figure 1a). Although the applied strain was high, the expected refinement of the two-phase microstructure was not observed. Within the phases, grain refinement was evident at *r*∼3 mm. Vickers microhardness measurements detected an increase of microhardness, (Figure 1c) and confirmed the ongoing refinement, which was seen in SEM within the phases. This observation indicated that the applied strain had acted in the composite to a certain degree. However, whether the total amount of applied strain had been absorbed by the composite through deformation was in question. The deformation at RT of samples with NiO of 50-at% (Figure 1a) or higher was considered to be unsuccessful because the samples had inhomogeneous microstructure and the FM-phase dimensions were too large for observing an H_eb_.

Deformation at elevated temperatures improved the deformation behaviour of the nanocomposites and led to a homogeneous microstructure (Figure 1b). SEM micrographs indicated that a homogeneous two-phase microstructure had been formed, having microstructural refinement from *r*∼0 mm to *r*∼3 mm. The large NiO structures, which were observed for T_def_ = RT, were not observed for T_def_ = 300 °C in the SEM micrographs along the sample radius. Here, the dark phase, which is shown in BSE-contrast, was determined by EDX measurements in combination with XRD-results (Figure 2) as Fe_3_O_4_ and the bright phase was identified as a FeNi-phase.

Vickers microhardness measurements detected a large difference in hardness between samples deformed at RT and 300 °C (Figure 1c). For the T_def_ = 300 °C composite, hardness values were considerably higher than for the RT sample, varying from 6 GPa at *r*∼0 mm (ϵ_vM_∼0) to 6.8 GPa at *r*∼3 mm (ϵ_vM_∼1740). Above *r*∼1.5 mm the T_def_ = 300 °C composite became microstructurally saturated according to Vickers microhardness measurements.

Further deformation experiments were conducted at deformation temperatures between RT and 300 °C (T_def_ = 200 °C, 225 °C, 250 °C and 275 °C) to gain insight into the onset of the microstructural homogenisation. Figure 1e displays SEM micrographs at *r*∼3 mm. At T_def_ = 200 °C, the microstructure was identical to that at RT; however, at T_def_ = 225 °C, the large NiO structures had become more and more refined through the application of strain. This process continued for T_def_ = 250 °C and T_def_ = 275 °C, gradually extending from the outer radius to the inner radius. At T_def_ = 275 °C, the microstructure is comparable to the T_def_ = 300 °C samples at *r*∼3 mm, but still large NiO structures were observed for T_def_ = 275 °C at r≤2 mm (not shown).

The obtained Vicker microhardness values reflected the observations from SEM. In Figure 1d hardness results at the outer radius are summarised. The inhomogeneous microstructure of the sample deformed at T_def_ = 200 °C had a lower microhardness (5 GPa) than those of samples deformed at temperatures T_def_ ≥ 225 °C, which were approximately 6 to 7 GPa at *r*∼3 mm and similar to the hardness values of the sample deformed at T_def_ = 300 °C. All samples processed at elevated temperatures exhibited significantly higher Vickers microhardness values than the sample processed at RT.

Phase analysis with XRD detected Fe and NiO phases for the samples deformed at RT and 200 °C with an onset of Fe_3_O_4_ formation at T_def_ = 200 °C (Figure 2). An increase of deformation temperature from T_def_ = 225 °C to T_def_ = 300 °C yielded a continuously growing amount of Fe_3_O_4_, caused by the reduction of NiO through Fe. Ni became available to build the γ-FeNi phase, which was detected in the XRD scans. The content of Fe within the γ-FeNi phase could be approximated through the shift to lower q-values. This shift is caused by the substitution of Fe-atoms within the Ni lattice, resulting in an enlargement of the lattice parameter. This enlargement is caused by the higher atomic radius of Fe compared to Ni [26]. The shifted peaks of γ-FeNi phase for T_def_ = 300 °C implied a composition of approximately γ-Fe_39_Ni_61_.

The quality of the XRD patterns and similarity of Fe_3_O_4_ and maghemite (γ-Fe_2_O_3_) patterns allowed a determination of Fe_x_O_y_ phase only at higher q_z_ ≥ 3.5 Å^−1^. The peaks at q_z_ = 3.889 Å^−1^ and q_z_ = 4.235 Å^−1^, for example, are in accordance with the Fe_3_O_4_ pattern. Presence of γ-Fe_2_O_3_ could not be excluded, but apparently Fe_3_O_4_ dominated the XRD-pattern; γ-Fe_2_O_3_ was therefore omitted in the following consideration for the Fe_50_NiO_50_ nanocomposite.

The assumption that the α-Fe phase had been completely consumed by the formation of Fe_3_O_4_ or γ-FeNi, allowed estimation of the residual NiO. Using the known composition of γ-Fe_39_Ni_61_ phase for the sample deformed at T_def_ = 300 °C, yielded a residual NiO-phase of maximum 16-wt%, which should have been left after synthesis inside the sample, but could not be resolved by the laboratory XRD-equipment due to overlapping XRD-peaks (Figure 2). Although the sample deformed at T_def_ = 300 °C had a promising microstructure, this nanocomposite was considered unsuitable for observing a significant H_eb_ in subsequent magnetometry measurements because it did not contain large amounts of NiO according to the XRD results.

The CSDS was calculated via Scherrer-relation of Fe_50_NiO_50_ sample at RT and T_def_ = 300 °C displayed nanocrystallinity, whereby Fe_3_O_4_ had the largest CSDS. Considering residual strain with the Williamson-Hall (WH) method [24] and comparing results, strain in the Fe_3_O_4_-phase is almost half compared to γ-FeNi. Strain-free CSDS estimations yielded approximately 45 nm for both analysed phases at T_def_ = 300 °C (Table 1).

#### 3.1.2. Fe_10_Ni_40_NiO_50_ Composition

To reduce the loss of NiO at 300 °C, another composition of Fe_10_Ni_40_NiO_50_ (Fe_8.4_Ni_35.4_NiO_56.2_ in wt%) was tested with the idea to conserve the NiO-phase by alloying Ni initially with Fe to prevent the reduction of NiO.

The microstructures depicted in Figure 3a were collected from two samples; one has been deformed with 30 rotations the other with 125 rotations; all other parameters were equal. For the micrographs shown, the applied strain was $\epsilon_{vM}$∼90 (≜*r*∼0.5 mm) and $\epsilon_{vM}$∼610 (≜*r*∼3.5 mm) for the sample with 30 rotations. For the sample with 125 rotations, the applied strain was equivalent to: $\epsilon_{vM}$∼270 (≜*r*∼0.375 mm), $\epsilon_{vM}$∼1280 (≜*r*∼1.75 mm) and $\epsilon_{vM}$∼2560 (≜*r*∼3.5 mm). Due to the homogeneous microstructure, a uniform deformation of the sample material was assumed and applied strain was therefore used rather than the radius. The dark phase was identified as NiO and the bright phase as Ni. The microstructure was different from the Fe_50_NiO_50_ sample deformed at T_def_ = 300 °C (for comparison Figure 1b). Through the preservation of NiO, a fragmentation process of NiO was realised, leading to a highly diverse Ni and NiO nanocomposite structure. Additionally, SEM micrographs demonstrated a refinement of the nanocomposite microstructure with an increase in applied strain.

Vickers microhardness measurements along the radial direction demonstrated a microstructural saturation at $\epsilon_{vM}$∼700 of 9.3 GPa in microhardness for the Fe_10_Ni_40_NiO_50_ sample after 125 rotations, (Figure 3b). Beyond $\epsilon_{vM}$∼700, further deformation caused only a small increase in microhardness of approximately 0.6 GPa to 9.9 GPa, which correlated with the minor refinement seen in SEM between $\epsilon_{vM}$∼1280 to $\epsilon_{vM}$∼2560. Vickers microhardness values of 9.3 to 9.9 GPa were significantly higher than previously reported results from similar nanocrystalline materials [27,28].

Synchrotron WAXS investigations were conducted along the axial direction for the following applied strains: $\epsilon_{vM}$∼270, $\epsilon_{vM}$∼1280, and $\epsilon_{vM}$∼2560. Detected phases were NiO, Ni and γ-Fe_2_O_3_. However, the γ-Fe_2_O_3_ peaks could not be directly identified. The comparison of WAXS patterns of the as-deformed and post-process annealed samples (Section 3.1.3) showed a slight shift from the Fe_3_O_4_ to the γ-Fe_2_O_3_ reference pattern (Figure 4). This shift became more evident in the calculated peak centres.

According to the WAXS measurements, the FM phase consisted of solely Ni. An expected substitution of Fe was thus not detectable with WAXS and implied that the main part of Fe was consumed by the formation of γ-Fe_2_O_3_. The WAXS results allowed estimation of the residual amount of NiO within the sample and yielded the following composition: Ni_48.6_NiO_39.4_ and (γ-Fe_2_O_3_)_12_, all in wt%.

Integral peak breadth analysation with the Scherrer-relation showed minor changes of CSDS for Ni and NiO, but the CSDS of γ-Fe_2_O_3_ reduced by 50% with applied strain (Table 2). The residual strain was approximated via the WH method and for NiO was approximately a factor of 10 higher than for Ni.

#### 3.1.3. Annealing of Fe_10_Ni_40_NiO_50_ Composition

Because of the complex microstructure of the nanocomposite after deformation, the effects of post-process annealing were investigated to understand the influence of phase growth and phase interface morphology on the magnetic properties of the Fe_10_Ni_40_NiO_50_ sample better. Hence, sample material with a large amount of applied strain $\epsilon_{vM}$∼1150 (*r*∼2.75 mm) was used to have.

The SEM micrographs are displayed in Figure 5 for the following states: as-deformed, annealed in vacuum at 450 °C and 550 °C. For the as-deformed state, the complex phase interface was assumed to have large variations in topography at a nanometre scale, which was expected to simplify through annealing. The contrast in BSE mode has been enhanced for the annealed samples. This enhancement indicated a phase growth of the bright and dark phases. The annealing led to a reduction of phase interface, which was expected to be predominantly driven by the optimization of Gibbs free energy.

XRD-peak analysation affirmed the rise of CSDS for all three detected phases (Table 2), whereby γ-Fe_2_O_3_ had the highest increase in relative crystallite size of approximately two times to its initial CSDS in the as-deformed state. The relative CSDS growth of Ni and NiO-phase during annealing was significantly smaller.

### 3.2. Magnetic Characterisation

For SQUID investigations, the promising microstructures of the Fe_10_Ni_40_NiO_50_ samples were chosen. The hysteresis loops are shown in Figure 6a have a continuously growing H_eb_, which could be correlated to the refinement of microstructure through the application of strain. The applied strain changed the hysteresis loop characteristic in general. As M_S_ decreased, coercivity rose. The coercivity nearly doubled from $\epsilon_{vM}$∼90 with H_C_ = 460 Oe to $\epsilon_{vM}$∼2560 having H_C_ = 842 Oe. Magnetic remanence (M_R_) rose from M_R_ = 18.5 emu/g to M_R_ = 23.3 emu/g at $\epsilon_{vM}$∼610 and became independent of the applied strain for further deformation. H_eb_ started at $\epsilon_{vM}$∼90 with H_eb_ = −52 Oe and ended at $\epsilon_{vM}$∼2560 with H_eb_ = −243 Oe (Table 3). As mentioned previously, M_S_ reduced from M_S,_$\epsilon_{vM}$∼90 = 37.2 emu/g to M_S,_$\epsilon_{vM}$ ∼2560 = 27.5 emu/g. These changes in M_S_ and H_eb_ appeared to be interrelated and might have had their origins in the detected changes in CSDS, enlargement of phase interfaces and the generation of γ-Fe_2_O_3_.

The slight increase in Vickers microhardness detected at *r* ≤ 1.25 mm (Figure 3b) was therefore confirmed through the continuously growing H_eb_ for such high amounts of applied strain and was in accordance with the impression of an ongoing refinement gained by SEM and XRD investigations.

Figure 6b shows hysteresis loops of the annealed samples at 8 K. The annealing treatment caused a drastic reduction of H_eb_ from H_eb,_$\epsilon_{vM}$∼1150 = −200 Oe to H_eb,ann. 550 °C_ = −23 Oe and M_R_ (Table 3), whereas M_S_ became larger, changing from M_S,_$\epsilon_{vM}$∼1150 = 32.3 emu/g to M_S,ann. 550°C_ = 36.6 emu/g. The rise of M_S_ with annealing treatment correlated with the detected growth of CSDS for all involved phases (Table 2).

The annealing initiated topological changes in the phase interface morphology, leading to a reduction of interface roughness. This would be considered to affect the H_eb_ as well [29,30], but its impact would be minor compared to the growth of phase size, especially in a polycrystalline sample [30]. The enlargement of FM-phase dimensions is qualitatively evident in the SEM micrographs shown in Figure 5 and is presumed to have been the main reason for the decline of H_eb_ by facilitating magnetic nucleation within the FM-phase, after annealing had been applied.

The detected change in H_eb_ versus applied strain for the Fe_10_Ni_40_NiO_50_ nanocomposite displayed a saturation behaviour similar to the microstructural saturation in single-phase systems previously reported for several phases deformed by HPT [18,31]. A simple fit model based on an asymptotic approximation yielded, for an asymptotic limit, an H_eb,max_ of 300 Oe (Figure 6c) for the Fe_10_Ni_40_NiO_50_ composite. The fit model is not based on a physical theory and was only used here to approximate an assumed saturation of H_eb_ through saturation of microstructural refinement.

## 4. Discussion

### 4.1. Deformation Behaviour

The deformation with HPT of composites containing ‘ductile-brittle’ phases has been reported for several phase systems including, Cu-Co and Cu-W [32,33]. For those systems, a fracturing of the ‘-brittle’ phase has been proposed to have a crucial influence on the refinement and homogenisation of the sample’s microstructure. Considering the Fe_50_NiO_50_ sample at RT, such fragmentation was not observed to an extensive degree. However, that does not imply that NiO cannot deform at RT. As shown in Figure 1a at RT the Fe and NiO deformed to lamella structures, but then reached a ’steady state’ without fragmentation of the NiO phase in reasonably large amounts. This effect is clear when the SEM micrographs at *r*∼0 mm and *r*∼3 mm are compared (Figure 1a). In contrast, Vickers microhardness increased with a radius for the sample deformed at RT, indicating an ongoing grain refinement within the phases (Figure 1c), which appeared to occur predominantly in the more ductile α-Fe phase.

With a body-centred cubic crystal structure, α-Fe has {110}<-111> as the primary slip system at RT. The {110}<-111> slip system family fulfils the von Mises criterion of five independent slip systems for the absorption of general strain via slip [34]. The mechanical properties of α-Fe are therefore different compared to ceramics such as NiO, which has a NaCl crystal structure with the primary slip systems {110}<1-10> at RT [34]. Basic consideration reveals, that such NaCl structures have two independent slip systems and that the von Mises criterion is not fulfilled; the absorption of plasticity by NiO is consequently limited. Hence, NiO cannot absorb general strain through slip, and the likelihood of the fragmentation of polycrystalline NiO is increased by the lack of matching slip systems of adjacent grains [34].

These considerations are important for a better understanding of the deformation process in such complex systems. Whether the initial phase is α-Fe or Ni, both phases are more ductile at RT than NiO. It could be that, especially at RT, the applied strain is absorbed by the α-Fe to a large extent, as the activation of slip is assumed to be easier in α-Fe than in the NiO-phase. It is plausible that through the elevated deformation temperatures, the activation of the additional slip systems is eased in NiO. The increased likelihood of slip in NiO could enhance fragmentation and consequently phase mixing. The eased activation of slip systems in NiO could be one reason for the drastic change of deformation behaviour at T_def_≥ 225 °C, which was seen for the Fe_50_NiO_50_ samples.

The discrete change of deformation behaviour between 200 °C and 225 °C (Figure 1e), could not be related to a specific transition temperature. Although NiO has a T_N_ of ∼251 °C [25], the expected change of plasticity around the T_N_ was not reported in [35]. The brittle-to-ductile transition temperature (BDTT) of NiO has been reported to be ∼0.3 of the melting temperature or approximately ∼600 °C [36], which is still significantly above 225 °C. In the same study, a brittle-to-ductile transition region was noted, ranging from 300 °C to ∼550 °C. Above 550 °C, NiO is considered to be ductile due to the activation of additional slip systems [36].

A further important point is the slip transfer across phase boundaries, which was observed in [37,38]. The likelihood for a glide or dislocation transfer between phases is higher, if the angle mismatch of adjacent slip systems is at a minimum [38], in addition to other conditions mentioned in [38]. HPT processing at elevated temperatures could lower the activation energy of additional slip systems within NiO. Therefore, the likelihood of dislocation transfer between adjacent phases is increased, because more slip systems are available inside NiO, if processed at elevated temperatures. This assumption could also explain the abrupt change in deformation behaviour observed in the Fe_50_NiO_50_ samples, when T_def_ was elevated from T_def_ = 200 °C to T_def_ = 225 °C.

Furthermore, the possibility cannot be excluded, that the application of high hydrostatic pressure reduced the activation energy of additional slip systems in NiO and caused a shift to lower BDTT. Activation of additional slip systems through hydrostatic pressure has been discussed in the literature [39,40] and has been demonstrated for an Mg-alloy, in which non-basal glide systems were activated with an applied pressure of ∼125 MPa [40].

Eased slip in NiO can lead to a roughening of interface morphology on a nanometre scale; this effect has been observed in other two-phase systems [37]. The absorption of dislocations from local pile-ups could cause local plastic instabilities, which are believed to support fragmentation of the more brittle phase, as discussed in a recent study [13].

When the Fe_10_Ni_40_NiO_50_ composition was deformed at T_def_ = 300 °C, the NiO-phase was conserved. In addition to the previously discussed impact of slip system activation and dislocation transfer in NiO, a work-hardening of the softer phase could also result in an enhanced fragmentation of NiO. Such a phenomenon has been reported for the Cu-Co-system during HPT-deformation [32]. The conservation of NiO facilitates a prolonged fragmentation, which could lead to the fine distribution of NiO embedded in Ni (Figure 3a).

NiO can break off from larger agglomerates, which would ease the reduction by Fe through the creation of additional phase interfaces. It is likely that the accumulation of γ-Fe_2_O_3_ occurs at phase interfaces or the grain boundary (GB). These small and dispersed clusters of γ-Fe_2_O_3_ could inhibit GB motion and accelerate grain refinement. In addition, through the efficient fragmentation of NiO, a highly diverse Ni and NiO-phase structure is created on a microscale, significantly hindering GB motion and plastic deformation. Both phenomena could have led to the small observed CSDS of approximately 20 nm according to the WH method at $\epsilon_{vM}$ > 1280 (Table 2) and could have caused the unusually high Vickers microhardness of 9 to 9.9 GPa.

The mismatch of slip systems and lattice parameters of participating phases, especially between γ-Fe_2_O_3_ and the Ni- and NiO phases, could have led to a dislocation pile-up at the γ-Fe_2_O_3_ phase, resulting in the creation of a non-crystalline region at the phase interface adjacent to γ-Fe_2_O_3_. Previous research has cited the build-up of a non-crystalline region at the phase interface of the Cu-Nb system as creating an impenetrable barrier for dislocation [41].

Comparing the observed Vickers microhardness values in Figure 3b to the results obtained for nanocrystalline Ni [27] or slightly oxidized Ni-powder processed by HPT [28] shows the influence of a nanostructured NiO phase on mechanical properties. Despite the reported high hardness values of approximately 7 GPa for nanocrystalline Ni or Ni with NiO clusters [27,28] the bulk nanocomposite of Fe_10_Ni_40_NiO_50_ surpassed these results by 30–40%.

### 4.2. Phase Formation during Deformation

The efficient reduction of NiO and the observed formation of Fe_3_O_4_ and γ-Fe_2_O_3_ for both investigated compositions was unanticipated. Commercially available Ni powder is reported to oxidize at >350 °C in ambient atmosphere [42]. In general, NiO is considered to be a stable oxide, and therefore a reduction of Fe was not expected. This reduction can be explained by comparing the heat of formation, which is significantly less for NiO than for Fe_3_O_4_ and γ-Fe_2_O_3_ [43,44]. The difference in heat of formation could also explain the one-way characteristic of the reaction.

According to the data, the onset of NiO reduction by Fe during HPT processing was initiated between 200 °C and 225 °C (Figure 2). This reaction occurred much more efficiently at T_def_ = 300 °C in the Fe_50_NiO_50_ material system. The formation of the thermodynamically more favourable Fe_3_O_4_ appears to have been enhanced by the extreme conditions during sample synthesis combined with the fragmented and nanocrystalline NiO, which provided a very large amount of interfacial phase area for the observed reduction of NiO. A slip in the {110}<1-10> system in the NiO phase could facilitate the oxidation of Fe because the occurrence of unsaturated oxygen bonds would be more likely. These two mentioned factors are considered to be the main cause for the observed two-phase system of Fe_3_O_4_ and γ-Fe_39_Ni_61_ for the Fe_50_NiO_50_ material system, when deformed at T_def_ = 300 °C.

NiO could not be resolved in the XRD scan for the Fe_50_NiO_50_ samples deformed at T_def_ = 300 °C (Figure 2), although 16-wt% NiO should have been left after synthesis inside the nanocomposite. The possibility that some residual NiO remained after deformation is supported by the analysis of the γ-FeNi phase XRD pattern. If complete reduction of NiO through Fe and complete incorporation of the residual Ni-atoms in the γ-FeNi phase is assumed, the composition of the γ-FeNi phase would change to γ-Fe_20_Ni_80_. The peak pattern of γ-Fe_20_Ni_80_ had its main peak at q_z_ = 3.073 Å^−1^ and was therefore distinguishable from the peak position of the observed γ-Fe_39_Ni_61_ phase.

The XRD results of Fe_50_NiO_50_ and Fe_10_Ni_40_NiO_50_ samples in Figure 2 and Figure 4, respectively, detected different formations of the Fe_x_O_y_ phase through the reduction of NiO. The formation of Fe_x_O_y_ is influenced by the amount of available O [45]. In the Fe_10_Ni_40_NiO_50_ sample, the Fe content was much lower compared to Fe_50_NiO_50_, and therefore sufficient O was available to form γ-Fe_2_O_3_.

Moreover, γ-Fe_2_O_3_ is thermodynamically more stable than Fe_3_O_4_ [43] and therefore is considered more likely to be formed, provided that sufficient O is available during deformation. Another difference from the Fe_50_NiO_50_ samples at T_def_ = 300 °C is the neglectable substitution of Fe inside the Ni. After the deformation of Fe_10_Ni_40_NiO_50_ composition, WAXS detected a γ-phase consisting solely of Ni.

### 4.3. Magnetic Properties

Regarding the magnetometry results, the aforementioned Fe-oxide phases (i.e., Fe_3_O_4_ and γ-Fe_2_O_3_) were both FM, but distinguished themselves in their magnetic properties, for example, bulk-M_S,Fe_3_O_4__ = 98.6 emu/g [46] and bulk-M_S,γ-Fe_2_O_3__ = 85.6 emu/g [47] at 8 K. Based on the results from WAXS measurements, the M_S,γ-Fe_2_O_3__ is used in the following discussion of the results for Fe_10_Ni_40_NiO_50_ nanocomposite.

The H_eb_ can be approximated for thin films with the simple theoretical model of $H_{eb}∼\sigma /(M_{S}\times t_{FM})$(σ = interfacial coupling energy density, t_FM_ = FM-thickness) [48]. Utilising this model, with the assumption that it is valid to some extent for nanocomposites measured at 8 K as well, allows estimation of the origin of the observed change of H_eb_ with applied strain.

The increase of H_eb,_$\epsilon_{vM}$∼90 = −52 Oe to H_eb,_$\epsilon_{vM}$∼2560 = −242 Oe cannot be explained by the decline of M_S_ alone. As shown in Table 3, M_S_ decreased by approximately 26% from M_S,_$\epsilon_{vM}$∼90 = 37.3 emu/g to M_S,_$\epsilon_{vM}$∼2560 = 27.6 emu/g, whereas H_eb_ increased almost fivefold from H_eb,min_ to H_eb,max_. Considering the observed changes of CSDS (Table 2) in addition to the decrease of M_S_ yields a more accurate estimation (Figure 7). The increase of H_eb,_$\epsilon_{vM}$∼270 = −90 Oe to H_eb,_$\epsilon_{vM}$∼2560 = −242 Oe can be explained to a great extent by the combined decreases of M_S_ and CSDS of γ-Fe_2_O_3_ using the approximation that σ was the same for both applied strains. This explanation implies, that NiO was exchange-biased to the γ-Fe_2_O_3_ as well as to the Ni-crystallites. From this calculation, it can be deduced that γ-Fe_2_O_3_ crystallites dominated the magnetic characteristic of the Fe_10_Ni_40_NiO_50_ samples and that the Ni-crystallites contributed partially.

The annealing experiments allow a deeper insight into the dependency of H_eb_ on the nanocomposite microstructure. The variable M_S_ recovered after deformation, from M_S,_$\epsilon_{vM}$∼1150 = 32.3 emu/g to M_S,ann. 550°C_ = 36.9 emu/g, and H_eb_ decreased to ∼10% of its initial value of H_eb,as def_ = −200 Oe through annealing at 550 °C. The annealed samples exhibited continuous CSDS growth for all three phases (Table 2) as well as phase size growth (Figure 5). Changes in CSDS in γ-Fe_2_O_3_ were approximately twofold as well, but the decrease of H_eb_ was more significant and could be described only partially by the previous estimation.

Although CSDS apparently had a crucial influence on H_eb_, magnetic properties in annealed samples can be influenced by other effects as well. The application of temperature can initialize optimisation of the phase interface area to lower surface energies and cause a reduction of phase interface topology. Those effects should also transform the morphology of the FM-AFM interface to a sharper phase gradient. The decreased interfacial area could affect the enclosure of FM grains by the AFM phase and reduce the overall magnetic stiffness inside the FM phase. Phase size growth could enable magnetic nucleation during field reversal, which could further weaken H_eb_.

The variation of M_S_ with applied strain or annealing temperature appears to have been related to some degree to the rise of CSDS of γ-Fe_2_O_3_, due to its large relative changes in size. Previous studies have found that FM-nanocrystallites possess a size-dependent M_S_ in general. The decline of M_S_ is related to a non-magnetic region at the crystallite boundary, which begins to influence M_S_ non-linearly below approximately 20 nm of crystallite dimension through a growing surface-to-volume ratio [49].

To obtain a better insight, the simple approximation from [49] was used for M_S_, based on the relationship $M_{S}(D_{Cry})=M_{S}(bulk)\times (1-3\times t_{non-mag}/D_{Cry})$(*D*_Cry_...crystallite size; for the calculation, the CSDSs from Table 2 and t_non-mag_...thickness of a non-magnetic layer between adjacent nanocrystallites were used) for nanocrystalline material. Utilising this model for the Fe_10_Ni_40_NiO_50_ nanocomposite, which consisted of 12-wt% γ-Fe_2_O_3_ and 48.6-wt% Ni; yields with a non-magnetic layer of 5.7 Å for γ-Fe_2_O_3_ [49], 3 Å for Ni (estimated for Ni) and assuming a bulk M_S,Ni_ = 58.6 emu/g [50] and M_S,γ-Fe_2_O_3__ = 85.6 emu/g [47]: M_cal.S,_$\epsilon_{vM}$∼270 = 35.7 emu/g, M_cal.S,_$\epsilon_{vM}$∼1280 = 35.1 emu/g and M_cal.S,_$\epsilon_{vM}$∼2560 = 34.2 emu/g, far more than measured (Table 3). These results suggest that the assumed thickness of the non-magnetic layer was underestimated for the considered nanocomposite and should have been much higher to match the experimental results. Regarding the different contributions to M_S_ of γ-Fe_2_O_3_ and Ni, crystallites that possess different CSDS allow the conclusion, that the decrease of M_S_ was mainly caused by the shrinking CSDS of γ-Fe_2_O_3_ crystallites through the deformation (Figure 8).

The results from the annealed samples support the previous argument, that the γ-Fe_2_O_3_ phase mainly influenced M_S_. Growth of γ-Fe_2_O_3_ from 8 nm to 14 nm CSDS would lead to the following calculated M_S_: M_cal.S,_$\epsilon_{vM}$∼1150 = 34.4 emu/g, M_cal.S,ann.450°C_ = 35.0 emu/g and M_cal.S,ann.550°C_ = 36.0 emu/g. The estimation for M_S,_$\epsilon_{vM}$∼1150 is certainly higher than the measured M_S_ value; however, M_S,ann.450 °C_ and M_S,ann.550 °C_ are in good accordance with the values presented in Table 3.

The discrepancy between M_S,_$\epsilon_{vM}$∼1150 and M_S,_$\epsilon_{vM}$∼2560 could be rooted in a larger number of non-magnetic regions than assumed. If complete oxidation of Fe to γ-Fe_2_O_3_ is assumed and crystal size effects are neglected, the calculated M_S_ for the annealed Fe_10_Ni_40_NiO_50_ sample yields, a value of M_S,cal_ = 38.3 emu/g. The M_S,ann. 550°C_ = 36.9 emu/g is strikingly similar to the theoretical calculation. This comparison suggests that the non-magnetic phase was predominantly composed of Ni and γ-Fe_2_O_3_ and was reduced to a minimum by the annealing treatment.

There have been reports of a phase transition from FM γ-Fe_2_O_3_ to the thermodynamically more favourable AFM α-Fe_2_O_3_ at temperatures of 370–600 °C [43]. While this transition could have occurred during annealing, an accumulation of non-crystalline regions through the application of strain is more likely. On one hand, the XRD peak pattern of α-Fe_2_O_3_ is distinct from γ-Fe_2_O_3_; on other hand, a formation of α-Fe_2_O_3_ cannot explain the rise in M_S_ through annealing. The plausible existence of non-crystalline regions at interfaces, presumably containing γ-Fe_2_O_3_, would have a fraction of bulk M_S_, M_S,amorphous_∼2–5% [51], and the transition from crystalline γ-Fe_2_O_3_ phase to a phase with non-crystalline regions at the crystallite boundary could explain the changes in M_S_, detected when large amounts of strain had been applied. Composites, such as Nb-Cu, deformed by SPD have been reported to contain non-crystalline regions at phase interfaces [41,52]. The crystallisation of these non-crystalline regions was observed when annealing was applied [41]. These results suggest that similar behaviour occurred in the Fe_10_Ni_40_NiO_50_ nanocomposite.

## 5. Conclusions

This study presents an insight into the deformation behaviour of phases possessing very distinct mechanical properties. Although NiO has, due to its NaCl-structure, different primary slip systems compared to Fe or FeNi, it was possible to synthesise nanocomposites with a homogeneous microstructure on the sub-micrometre regime. The importance of thermal HPT processing was demonstrated on the Fe_50_NiO_50_ composition. Through the deformation at 300 °C, a homogeneous two-phase microstructure evolved. XRD investigations confirmed a reduction of NiO through Fe leading to the formation of Fe_3_O_4_ and γ-Fe_39_Ni_61_.

To conserve the AFM NiO in respect to enhance the H_eb_, a composition of Fe_10_Ni_40_NiO_50_ was synthesised by HPT at 300 °C and nanocomposites with a unique microstructure were obtained. SEM micrographs showed a very homogeneous microstructure, which primarily contained NiO and Ni according to WAXS investigations. The formation of an additional phase was detected by WAXS, which consisted of γ-Fe_2_O_3_. Vickers microhardness measurements detected a microstructural saturation behaviour with a slight increase to outer radii to an unusually high Vickers microhardness of ∼9.9 GPa. WAXS, SEM and magnetometry results conveyed an ongoing refinement of the nanocomposite’s microstructure.

It was possible to demonstrate the existence of H_eb_ in such bulk-sized nanocomposite and the continual increase of H_eb_ with applied strain was correlated to the evolving microstructure. The rise of H_eb_ was mainly attributed to a simultaneous decrease in M_S_, reduction of FM-phase size dimensions and crystallite size of the FM-phases, predominantly γ-Fe_2_O_3_.

Annealing experiments of Fe_10_Ni_40_NiO_50_ samples showed a decrease of H_eb_ and an increase of M_S_. Those changes of H_eb_ and M_S_ contributed to the growth of FM-phase dimensions and the subsequent reduction of phase interfaces.

## Figures and Tables

**Figure 1 nanomaterials-13-00344-f001:**
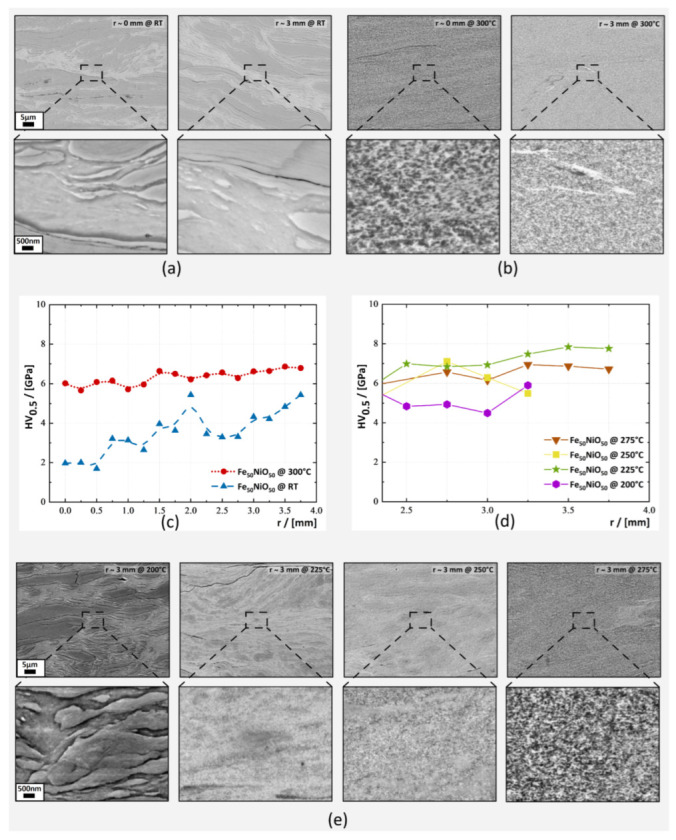
SEM micrographs in BSE mode depict samples deformed at RT (**a**) and 300 °C (**b**) at r=0 mm and r=3 mm. EDX spectroscopy identified the dark phase as NiO for (**a**) or Fe_3_O_4_ for (**b**). (**c**) Vickers microhardness of Fe_50_NiO_50_ samples along the radial direction. The sample processed at 300 °C had a higher saturation microhardness compared to the sample deformed at RT. (**d**) Vickers microhardness values form r=2.5 mm to r=3.75 mm. Higher deformation temperatures led to a homogeneous microstructure and therefore higher hardness values. (**e**) SEM micrographs in BSE mode gained of Fe_50_NiO_50_ samples at the outer radii (*r*∼3 mm). An onset of homogenisation of microstructure could be seen at 225 °C and gradually improved towards 275 °C. The micron bar applies to all micrographs in the same row.

**Figure 2 nanomaterials-13-00344-f002:**
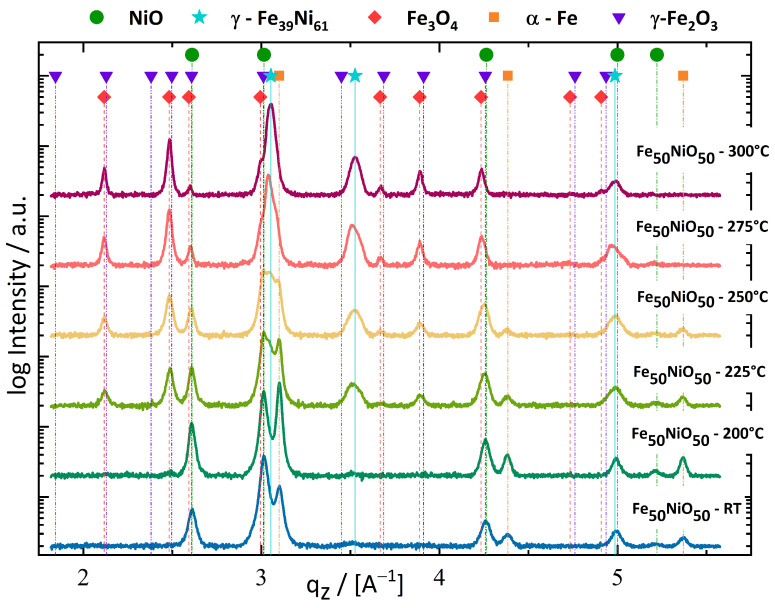
XRD-data is displayed of the Fe_50_NiO_50_ sample, which was measured with laboratory equipment. The formation of Fe_3_O_4_, through the reduction of NiO by Fe, was seen for samples deformed at T_def_ ≥ 225 °C. α-Fe was not detectable at T_def_ ≥ 275 °C.

**Figure 3 nanomaterials-13-00344-f003:**
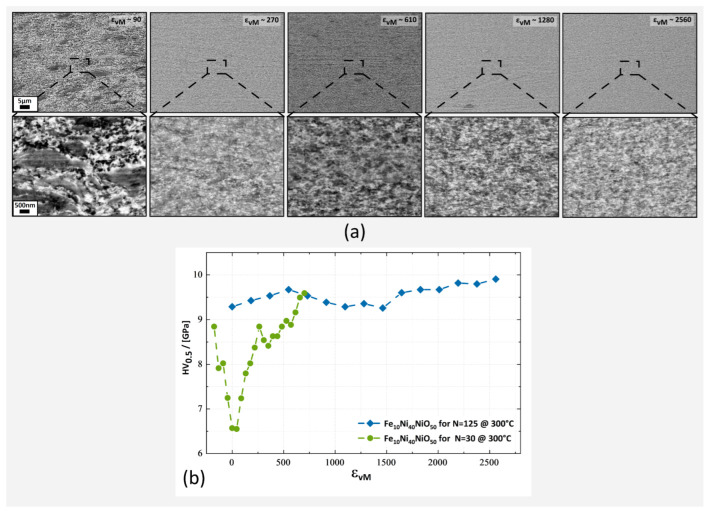
(**a**) BSE micrographs of the Fe_10_Ni_40_NiO_50_ samples. Microstructural refinement was detected from left to right. Two samples were deformed to obtain the depicted micrographs. First samples were deformed with 30 rotations and micrographs of an applied strain of $\epsilon_{vM}$∼90 to $\epsilon_{vM}$∼610 were taken. The second samples were deformed with 125 rotations and micrographs of an applied strain of $\epsilon_{vM}$∼270, $\epsilon_{vM}$∼1280 and $\epsilon_{vM}$∼2560 were taken. The micron bar applies to all micrographs in the same row. (**b**) Vickers microhardness of both Fe_10_Ni_40_NiO_50_ samples deformed at 300 °C.

**Figure 4 nanomaterials-13-00344-f004:**
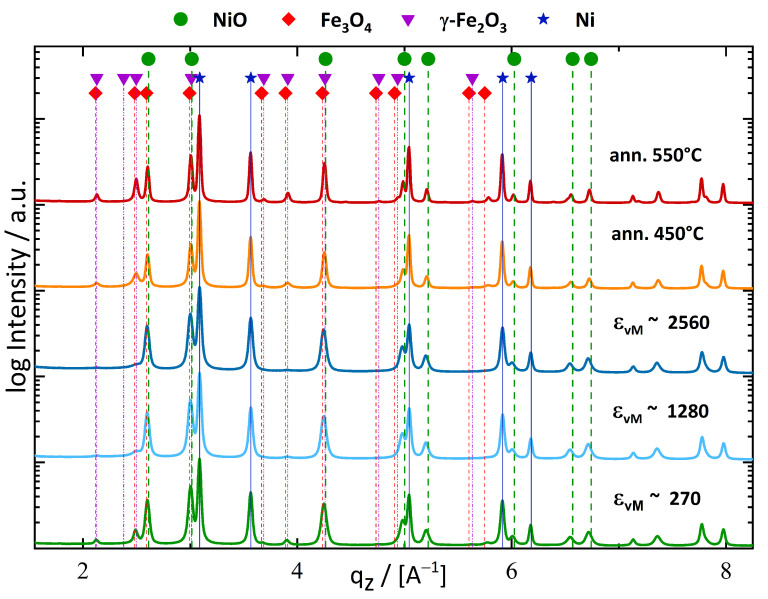
Synchrotron WAXS results are shown for different applied strains of the Fe_10_Ni_40_NiO_50_ sample. Starting from bottom: $\epsilon_{vM}$∼270, $\epsilon_{vM}$∼1280 and $\epsilon_{vM}$∼2560. The two annealed samples are at the top: atop the sample annealed at 550 °C and below the sample annealed at 450 °C. At low q-values the development of γ-Fe_2_O_3_ peaks were visible. γ-Fe_2_O_3_-peaks became weaker with high applied strains due to a reduction of CSDS.

**Figure 5 nanomaterials-13-00344-f005:**
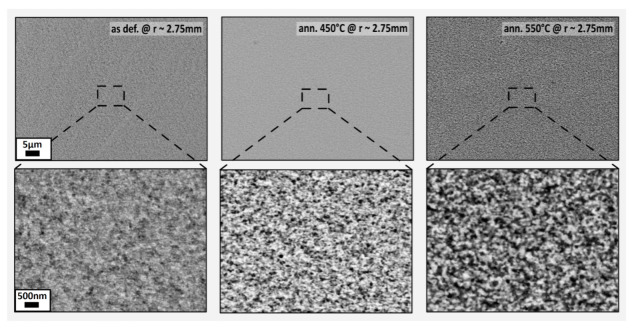
SEM micrographs were done in BSE mode of as-deformed and annealed samples of the Fe_10_Ni_40_NiO_50_ composition. The annealing initiated a phase growth and change of interface morphology for the samples annealed at 450 °C and 550 °C. The micron bar applies to all micrographs in the same row.

**Figure 6 nanomaterials-13-00344-f006:**
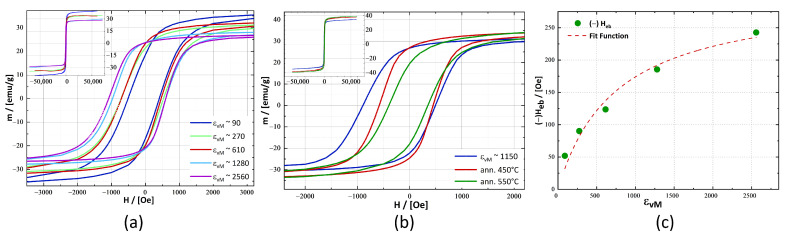
The magnetic hysteresis loops of the Fe_10_Ni_40_NiO_50_ sample were measured at 8 K. The graph displays a zoom-in at ±2000 Oe. The inlet in the left corner provides the whole hysteresis loop. (**a**) Hysteresis loop of as-deformed samples (**b**) Hysteresis loops of the annealed samples. (**c**) H_eb_ for different amounts of applied strain. The used fit is not based on a physical model and serves only to approximate the expected saturation of H_eb_ for an infinite amount of applied strain.

**Figure 7 nanomaterials-13-00344-f007:**
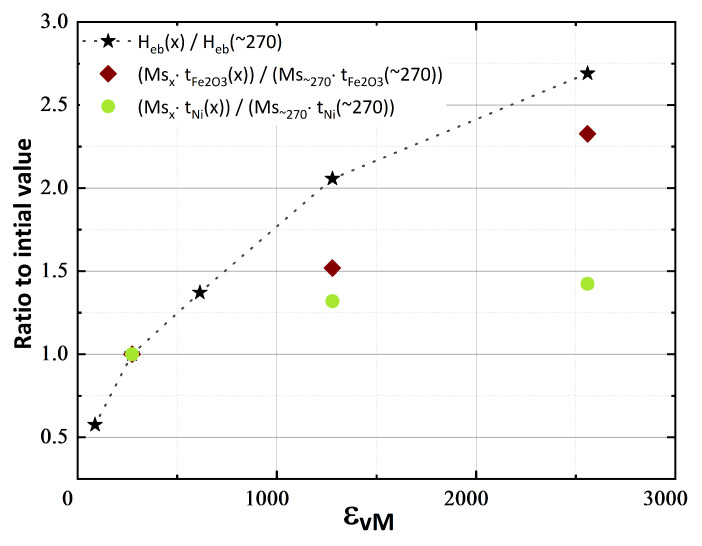
Relative change of H_eb_ related to H_eb,_$\epsilon_{vM}$∼270 and comparison of the ratio to the relative changes of M_S_ and CSDS size of t_γ-Fe_2_O_3__ and t_Ni_.

**Figure 8 nanomaterials-13-00344-f008:**
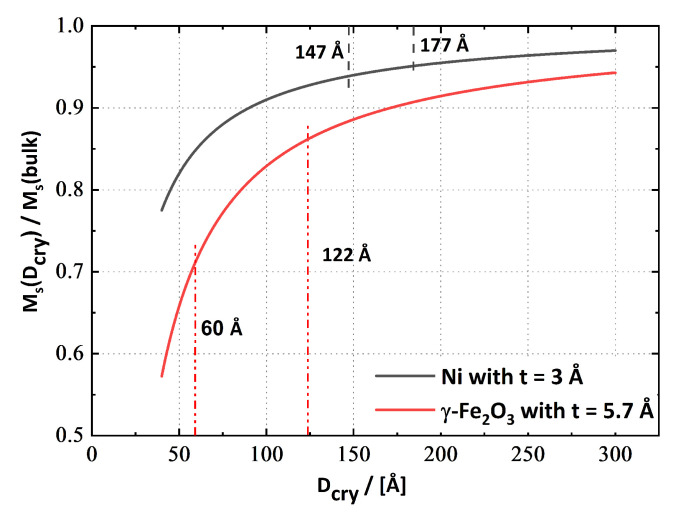
The influence of crystallite size on M_S_, according to [49], assuming a non-magnetic layer between adjacent crystallites of 5.7 Å for γ-Fe_2_O_3_ and 3 Å for Ni. Due to the smaller CSDS for γ-Fe_2_O_3_ the surface-to-volume ratio becomes larger, causing a stronger decrease of M_Sγ-Fe_2_O_3__.

**Table 1 nanomaterials-13-00344-t001:** Summary of peak analysis executed for Fe_50_NiO_50_ sample deformed at RT and 300 °C. <D_x_> is the average crystallite size or CSDS, D_x-WH_ represents the strain-less CSDS result and ϵ_x-WH_ residual strain in the crystallite, both values are obtained via the WH method.

	<D_Fe_> /[nm]	D_Fe-WH_/[nm]	ϵ _Fe-WH_	<D_NiO_ > /[nm]	D_NiO-WH_/[nm]	ϵ _NiO-WH_
Fe_50_NiO_50_ @T_def_ = RT	11 ± 2	40	2 × 10${}^{-2}$	11 ± 2	30	2 × 10${}^{-2}$
	**<D_FeNi_> / [nm]**	**D_FeNi-WH_/[nm]**	** ϵ _FeNi-WH_ **	**<D_Fe_3_O_4__>** **/[nm]**	**D_Fe_3_O_4_-WH_/[nm]**	** ϵ _Fe_3_O_4_-WH_ **
Fe_50_NiO_50_ @T_def_ = 300 °C	12 ± 2	45	3 × 10${}^{-2}$	19 ± 7	43	1 × 10${}^{-2}$

**Table 2 nanomaterials-13-00344-t002:** A summary of integral peak breadth analysis is shown, which was done for the Fe_10_Ni_40_NiO_50_ sample deformed at T_def_ = 300 °C. <D_x_> is the average crystallite size or CSDS, D_x-WH_ represents the strain-less CSDS result and ϵ_x-WH_ residual strain in the crystallite both values were obtained by the WH method.

	<D_Ni_> /[nm]	D_Ni-WH_/[nm]	ϵ _Ni-WH_	<D_NiO_> /[nm]	D_NiO-WH_/[nm]	ϵ _NiO-WH_	<D_γ-Fe2O3_> /[nm]
ϵ_vM_∼270	18 ± 2	21	3 × 10${}^{-3}$	11 ± 3	25	2 × 10${}^{-2}$	12 ± 2
ϵ_vM_∼1280	15 ± 5	19	2 × 10${}^{-3}$	11 ± 2	20	1 × 10${}^{-2}$	9 ± 4
ϵ_vM_∼2560	15 ± 4	18	2 × 10${}^{-3}$	11 ± 2	19	1 × 10${}^{-2}$	6 ± 2
ann. 450 °C	21 ± 2	22	1 × 10${}^{-3}$	14 ± 3	22	9 × 10${}^{-3}$	8 ± 2
ann. 550 °C	24 ± 2	25	0.5 × 10${}^{-3}$	17 ± 3	26	7 × 10${}^{-3}$	14 ± 2

**Table 3 nanomaterials-13-00344-t003:** Summary of magnetic hysteresis loops of Fe_10_Ni_40_NiO_50_ samples measured at 8 K, which are shown in Figure 6a,b. Exchange bias (H_eb_), symmetric coercivity (H_C_), coercivity center side (H_C1_), coercivity right side (H_C2_), saturation magnetisation (M_S_) and magnetic remanence (M_R_).

	H_eb_/[Oe]	H_C_/[Oe]	H_C1_/[Oe]	H_C2_/[Oe]	M_S_/[emu/g]	M_R_/[emu/g]
ϵ_vM_∼90	−52	460	−512	408	37.3	18.6
ϵ_vM_∼270	−90	625	−715	535	32.6	21.2
ϵ_vM_∼610	−124	587	−710	463	34.1	23.3
ϵ_vM_∼1280	−186	786	−971	600	28.8	23.7
ϵ_vM_∼2560	−243	843	−1085	599	27.6	23.7
ϵ_vM_∼ 1150	−200	697	−894	494	32.3	26.8
ann. 450 °C	−67	515	−581	449	35.8	27.0
ann. 550 °C	−23	373	−396	349	36.6	19.6

## Data Availability

Not applicable.

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
