# Peer review of "Exchange Bias Demonstrated in Bulk Nanocomposites Processed by High-Pressure Torsion"

_nanomaterials, 2023, doi:10.3390/nano13020344_

Round 1
Reviewer 1 Report
The manuscript reports on the preparation of Fe/Ni/NiO nanocomposites, followed by morphology, structural and magnetic characterization. The manuscript is clear in data presentation, and conclusions are fully supported by the previous discussion of magnetic data in light of structural features and preparation procedures. The data are original and very interesting for nanomaterials readers. For these reasons, the manuscript can be accepted after fixing the following minor notes:
11) Section 3.1.1 at the begininng: “at r ∼0 mm and r ∼3 mm.”: It is not clear must be explained better this nomenclature of the samples. Especially “r ∼0 mm” requires explanation
22) Section 3.1.2 at the beginning: “hinder” is not the correct term, please rewrite
33) Section 4.1 line 6: “reache” please correct
44) 15 lines later “crystal following [35].”: not clear, please rewrite
Author Response
Open comments from reviewer 1:
11) Section 3.1.1 at the begininng: “at r ∼0 mm and r ∼3 mm.”: It is not clear must be explained better this nomenclature of the samples. Especially “r ∼0 mm” requires explanation
I address this point now in the material and methods section. I think it is a very important point, thank you for your comment.
The description is now in 2. Materials and methods line 97 to 101 in the pdf
"In order to assign the corresponding radial position to the SEM images, the ends of the examined HPT half discs were determined during the SEM examination. r ∼0 mm was defined as the half distance between both ends, and r ∼3 mm was determined by the relative distance to r ∼0 mm. r ∼3 mm was cross-checked by measuring the relative distance to the end of the disc, which should be 1 mm."
22) Section 3.1.2 at the beginning: “hinder” is not the correct term, please rewrite
is corrected: line 222 in the pdf
"was used with the idea to conserve NiO-phase by alloying Ni initially with Fe to prevent the reduction of NiO. "
33) Section 4.1 line 6: “reache” please correct
corrected: line 328 in the pdf
but then reaches a ’steady state’ without fragmentation
44) 15 lines later “crystal following [35].”: not clear, please rewrite
corrected line 334 to 336in the pdf
"Hence, NiO cannot absorb general strain through slip and the likelihood for the fragmentation of polycrystalline NiO is increased by the lack of matching slip systems of adjacent grains [37]."
Reviewer 2 Report
The introduction should be improved, to find out what is the problem, what is new in this work, instead of what has been done.
Author Response
Comment of Reviewer 2:
The introduction should be improved, to find out what is the problem, what is new in this work, instead of what has been done.
Thank you for your comment: The introduction was changed to underline more what is new in the current report, especially about the Heb and the investigation in 3D samples.
Reviewer 3 Report
Ferromagnetic (Fe or Fe20Ni80) and antiferromagnetic (NiO) phases were deformed by high pressure torsion, a severe plastic deformation technique, to manufacture bulk sized nanocomposites and demonstrating an exchange bias, which was up to now predominantly reported for bilayer thin film. In this paper, HPT deformation was used to process a FM (Fe or Fe20Ni80) with an AFM NiO phase to obtain bulk nanocomposites. High pressure torsion deformation at elevated temperatures proved to be critically to obtain homogeneous nanocomposites. Additionally, a tailoring of magnetic parameters was demonstrated by the application of different strains or post-process annealing. This work can give some guidance to manufacture bulk sized nanocomposites. However, before considering publication, there are some issues that need to be clarified. The detailed comments are as follows:
1. The graphical abstract needs to be added.
2. Please describe the application of nanomaterials in recent years in the introduction. Nanocrystalline materials have drawn a great deal of attention in recent decades [1, 2]. Some new materials references should be added. Such as Environmental Functional Materials 1 (2022) 166-181; Journal of Catalysis, 2022, 415, 218-235; Journal of Catalysis 417 (2023) 116–128
3. Why did you choose FM (Fe or Fe20Ni80) nanocomposites with an AFM NiO phase? And please describe its advantages in the introduction.
4. Please confirm that the abscissa name format in Figure 6c is correct.
5. In the seventh paragraph of 4.1, (see Figure 3 a) should be changed to (Figure 3 a).
6. Is it necessary to draw the material synthesis steps and annealing experiment steps?
7. Some tense problems, grammatical problems and the beauty of pictures in the article need to be further improve.
Author Response
Points to revise from Reviewer 3:
1.) A graphical abstract was added to the revised material
2.) I was discussing your suggested papers, but we considered them as not suitable for the scope of this article. I implemented, as you suggested, some application with the focus on material science and additionally a recent article "Progress in Materials Science 2018, 94, 462–540". From my point of view, if I added the application in catalysis, I would also have to include the applications in medicine and so on, and this was considered to be far beyond the scope of this current article.
The application and importance of nanocrystalline material in the field of material science are mentioned in line 21 to 41 in the new pdf
3.) A proper description is now implemented in the introduction, why the specific materials were chosen. line 41to 51 in the new pdf
4.) Figure 6.c) the name of the abscissa is applied strain εvM. The size was increased
5.) Thank you, it is corrected
6.) From my point of view, it is not necessary, especially for the post-process annealing. The process of powder compaction is simple and straight forward. In the graphical abstract, an illustration was implemented for the powder compaction, but without the airtight capsule.
Further in 2. Materials and Methods line 82 to 86 in the pdf some information was added: "Powders were stored and prepared inside a glove box filled with Ar-atmosphere. For powder consolidation with HPT an airtight capsule was used to prevent the powder from oxidation. The capsule itself enclosed the HPT anvils, which can move freely only in axial direction for powder blend compaction. The obtained pellet was used later on in a second step for HPT deformation."
7.) I tried to address your comment accordingly, but due to the holidays, all our native speakers at the institute left already for their families before I could ask them for a proper proofreading.
At the first submission my Latex spellchecker was not working, I could fix the problem and I found several typos. I also tried to revise the grammar as good as I can.
Round 2
Reviewer 3 Report
-
The paper is insufficiently revised
Please reply to the comments one by one and revise the paper according to the comments.
Author Response
Please find the point to point revision of nanomaterials-2116464 in the attached Word file: Revision_02_nanomaterials-2116464

Round 3
Reviewer 3 Report
The paper cites a large number of references from a long time ago, but there are few references in the last five years, which are mainly concentrated in the first few introductions, without substantial references. The innovation and research significance of the paper cannot be seen through the quoted literatures, so it is difficult for the paper to be approved
Author Response
Dear Reviewer, please find attached the responses to your comments in the file:
